# Nickel-catalyzed asymmetric hydrogenation for the preparation of α-substituted propionic acids

Bowen Li [1,3], Zhiling Wang[1,3], Yicong Luo[1], Hanlin Wei[1], Jianzhong Chen [1] ✉, Delong Liu[2] & Wanbin Zhang [1,2] ✉

Transition metal-catalyzed asymmetric hydrogenation is one of the most efficient methods for the preparation of chiral α-substituted propionic acids. However, research on this method, employing cleaner earth-abundant metal catalysts, is still insufficient in both academic and industrial contexts. Herein, we report an efficient nickel-catalyzed asymmetric hydrogenation of α-substituted acrylic acids affording the corresponding chiral α-substituted propionic acids with up to 99.4% ee (enantiomeric excess) and 10,000 S/C (substrate/catalyst). In particular, this method can be used to obtain (R)-dihydroartemisinic acid with 99.8:0.2 dr (diastereomeric ratio) and 5000 S/C, which is an essential intermediate for the preparation of the antimalarial drug Artemisinin. The reaction mechanism has been investigated via experiments and DFT (Density Functional Theory) calculations, which indicate that the protonolysis of the C-Ni bond of the key intermediate via an intramolecular proton transfer from the carboxylic acid group of the substrate, is the rate-determining step.

Optically pure α-substituted propionic acids are widely used as biologically active compounds and important organic intermediates[1–6] (Fig. 1a). For instance, chiral α-aryl substituted propionic acids represent a very important class of anti-inflammatory and analgesic agents, such as (S)-Ibuprofen, (S)-Naproxen, (S)-Flurbiprofen, and (S)-Ketoprofen, because they are easily absorbed by organisms and have the effect of inhibiting the synthesis of prostaglandins[1]. Also, there are many other useful molecules, such as chiral α-aryl substituted propionate esters, which have been developed through simple derivations of chiral α-aryl substituted propionic acids. For example, chiral α-aryl substituted propionate esters can be used as an anti-tumor candidate drug[2] and potent inhibitors against the inflammatory phenotype of cystic fibrosis[3]. Additionally, the chiral α-alkyl substituted propionic acid derivatives can also serve as important drugs and synthetic intermediates. Two representative examples are the world-renowned antimalarial drug Artemisinin[4] and

the widely used chiral intermediate, (S)-Roche ester[5]. Given the very important applications of these compounds, their asymmetric synthesis has long been a prominent research topic for chemists. Among the reported methodologies, asymmetric hydrogenation, as an efficient and easy-to-industrialize preparation method, is one of the most attractive approaches[7–35] (Fig. 1b).

In the pioneering work on asymmetric hydrogenation, Prof. Knowles discovered that Rh complexes can catalyze the asymmetric hydrogenation of α-substituted acrylic acids[20]. Although the corresponding product was obtained in only 15% ee, this work opened the door to the study of transition metal-catalyzed asymmetric hydrogenation of these types of substrates catalyzed by rhodium complexes, affording the desired products in up to 99% ee and

[1]Shanghai Key Laboratory for Molecular Engineering of Chiral Drugs, Frontiers Science Center for Transformative Molecules, School of Chemistry and Chemical Engineering, Shanghai Jiao Tong University, 800 Dongchuan Road, Shanghai 200240, China. [2]School of Pharmacy, Shanghai Jiao Tong University, 800 Dongchuan Road, Shanghai 200240, China. [3]These authors contributed equally: Bowen Li, Zhiling Wang. ✉e-mail: 0091109001@sjtu.edu.cn; wanbin@sjtu.edu.cn

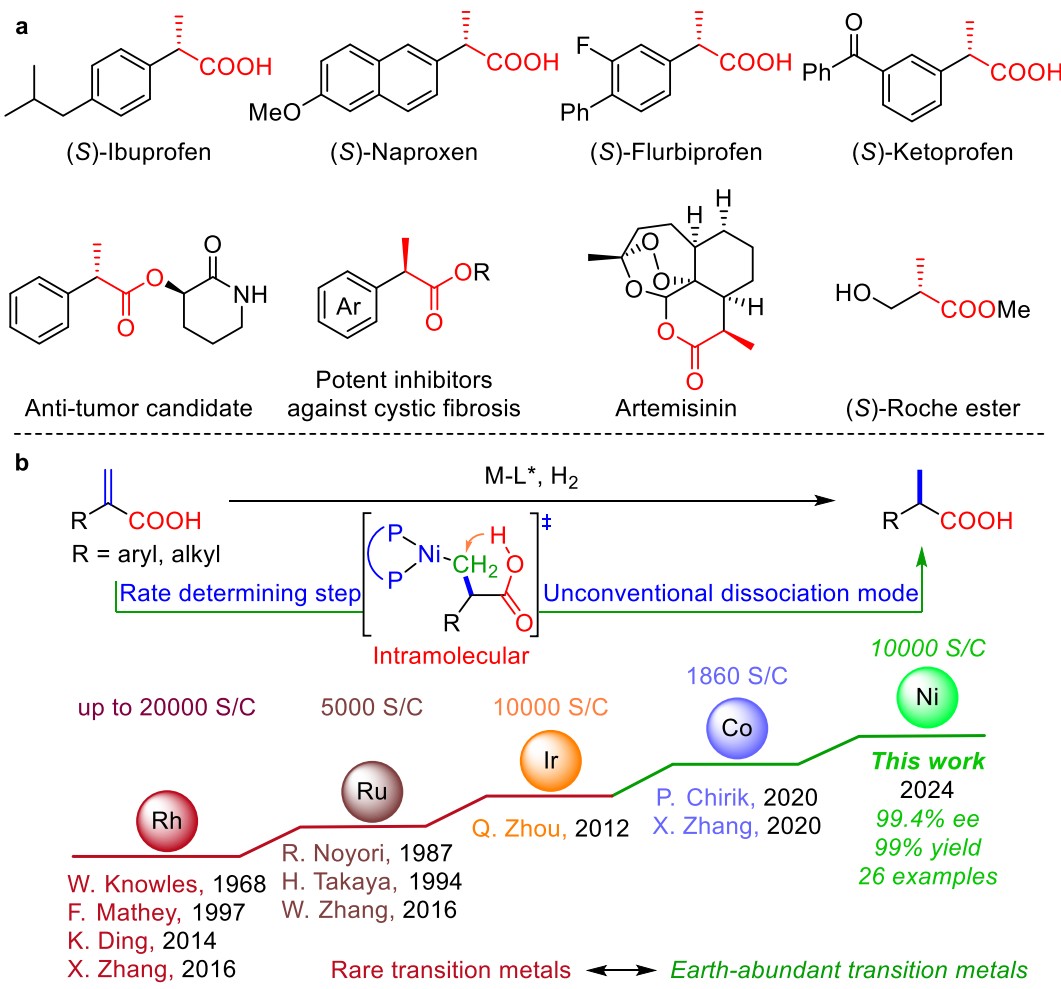

**Fig. 1 | The study on asymmetric hydrogenation of α-substituted acrylic acids. a** Representative molecules bearing chiral α-substituted propionic acid scaffolds. **b** Transition metal-catalyzed asymmetric hydrogenation of α-substituted acrylic acids.

20000 S/C[23]. In addition, in 1987 and 2012, two other rare metals ruthenium and iridium hydrogenation catalysts for the preparation of chiral α-substituted propionic acids were successively developed at first by Profs. Noyori[25] and Zhou[28] respectively (Fig. 1b).

In recent years, asymmetric hydrogenation catalyzed by earth-abundant metals (such as Mn, Fe, Co, Ni, and Cu) has made significant progress[36–49]. Among them, The latest reports by Profs. Chirik[29] and Zhang[30] on the cobalt-catalyzed asymmetric hydrogenation of α,β-unsaturated carboxylic acids (including α-substituted acrylic acids) in 2020 represent two excellent examples (Fig. 1b). Recently, nickel catalysts have received much attention when employed in asymmetric hydrogenation, particularly by Hamada[50,51], Zhou[52–55], Chirik[56], Zhang[57–61], our group[62–66], and other research groups[67–72]. However, perhaps owing to the lower steric hindrance of disubstituted olefins, which complicates the control of stereoselectivity in the reaction, the majority of studies have concentrated on tri- and tetra-substituted olefins[50–72]. In recent years, we have discovered that the multiple attractive dispersion interactions (MADI) between catalyst and substrate have a great influence on the activity and selectivity of asymmetric catalytic reactions[49,62–66]. Thus, we envisioned whether this discovery could be applied to the asymmetric hydrogenation of such disubstituted substrates. Herein, we report an efficient enantioselective nickel-catalyzed hydrogenation of α-substituted acrylic acids to provide the corresponding chiral α-substituted propionic acid products with excellent results. The efficient hydrogenation (10,000 S/C) proceeds through an unusual process of protonolysis of the C–Ni bond.

This involves intramolecular proton transfer from the carboxylic acid group of the substrate, to release the product and regenerate the catalyst, which is different from the reported dissociated methods using hydrogen or solvent hydrogen cations (Fig. 1b).

## Results

### Investigation of reaction conditions

Initially, 2-phenylacrylic acid (**1a**) was chosen as a model substrate for asymmetric hydrogenation using 1.0 mol% Ni(OAc)₂·4H₂O and *P*-chiral (*R*,*R*)-QuinoxP* with large electron-donating groups, which can generally improve the activities and enantioselectivities of metal catalysts[62]. The reaction was conducted under 30 bar H₂ at 50 °C in TFE (2,2,2-trifluoroethanol) over 24 h. As a result, the reaction provided a moderate conversion and enantioselectivity (Table 1, entry 1, 70% conv., 76% ee). To our delight, another similar *P*-chiral ligand, (*R*,*R*)-BenzP*, showed excellent reactivity and enantioselectivity (entry 2, >99% conv., 96% ee). However, only 17% conversion was obtained using (*R*)-BINAP as the ligand (entry 3). Next, when several other commonly used chiral diphosphine ligands and other (single or mixed) solvents were tested, no improvement in this hydrogenation was observed (see Supplementary Table 1 for details). By lowering the temperature to 30 °C, **2a** could be obtained with 98% ee, but the reaction did not proceed to completion (entry 4). When the reaction temperature is 30 °C and the H₂ pressure is 50 bar, the reaction proceeded with 97% conversion and 97% ee (entry 5). In order to test the catalytic efficiency, we reduced the catalyst loading to 0.2 mol% (S/C = 500), and substrate

## Table 1 | Reaction optimization[a]

| entry | ligand | temp. (°C) | conv. (%)[b] | ee (%)[c] |
|---|---|---|---|---|
| 1 | (R,R)-QuinoxP* | 50 | 70 | 76 |
| 2 | (R,R)-BenzP* | 50 | >99 | 96 |
| 3 | (R)-BINAP | 50 | 17 | – |
| 4 | (R,R)-BenzP* | 30 | 93 | 98 |
| 5[d] | (R,R)-BenzP* | 30 | 97 | 97 |
| 6[e] | (R,R)-BenzP* | 50 | >99 | 96 |

(R,R)-QuinoxP*          (R,R)-BenzP*          (R)-BINAP

[a]Reaction conditions: **1a** (0.20 mmol), Ni(OAc)$_2$·4H$_2$O/ligand (1.0 mol%, S/C = 100), TFE (1.0 mL), H$_2$ (30 bar), 50 °C, 24 h. [b]The conversions were calculated from $^1$H NMR spectra. [c]The ee values were determined by HPLC using chiral columns. [d] 50 bar H$_2$. [e] **1a** (0.50 mmol), Ni(OAc)$_2$·4H$_2$O/(R,R)-BenzP* (S/C = 500), TFE (2.0 mL).

**1a** could still be completely converted to its corresponding product (entry 6). After screening nickel salts (see Supplementary Table 1 for details), the use of 0.20 mol% Ni(OAc)$_2$·4H$_2$O and (R,R)-BenzP* under 30 bar of H$_2$ at 50 °C in TFE was selected as the optimal reaction conditions.

### Scope of asymmetric catalysis of α-substituted acrylic acids

Under the optimized reaction conditions, the substrate scope of the α-substituted acrylic acids **1** was explored (Fig. 2). All substrates provided the corresponding products with full conversions and excellent enantioselectivities (90-99.4% ees), with only a few substrates requiring a slight reduction in S/C (5 examples with 250 S/C and 2 examples with 100 S/C) perhaps due to their low solubility or slightly poor activity. When the substituents are located at the *ortho*-position of the aryl groups (**1b-f**), the substrates provided the corresponding products (**2b−f**) with better enantioselectivities (97-99.4% ees) than **2a** (96% ee). The aryl acrylic acids containing *meta*- (**1g, 1h**) and *para*-substituents (**1i−n**) also exhibited excellent stereoselectivities in this hydrogenation (92−96% ees). Then, the substrates bearing disubstituted aryl were also explored. The several typical substrates (**1o-s**) provided good catalytic results (92−99.2% ees). Next, a range of chain and cyclic alkyl substrates (**1t-y**) proceeded smoothly and provided the corresponding hydrogenation products with excellent enantioselectivities (90-95% ees). In addition, some heteroaromatic and trisubstituted substrates are not suitable for this catalytic system (see Supplementary Fig. 1 for details). The absolute configuration of product **2f** was assigned to be *R* by X-ray crystallographic analysis (see Supplementary Fig. 8 for details).

### The study of catalyst efficiency and synthetic applications

To further evaluate the activity of the catalyst and applicability of this catalytic system, the catalyst loading was first tested. To our delight, the model substrate **1a**, in the presence of a much lower catalyst loading (1/10000), was reacted completely on a gram scale to give **2a** with 98% yield and 96% ee, albeit requiring a little longer to complete (Fig. 3a). To the best of our knowledge, this result represents the highest TON (turnover number) for the Ni- catalyzed asymmetric hydrogenation of olefins reported to date[36–72]. The product can also be further transformed into an anti-tumor candidate[2] and potent

inhibitors against an inflammatory phenotype[3] (Fig. 3a) according to literature procedures. Dihydroartemisinic acid (R)-**2z** is the key intermediate for preparing Artemisinin, which is one of the most effective drugs for the treatment of malaria[4]. Thus, the asymmetric hydrogenation of artemisinic acid (**1z**) was conducted using this catalytic system. Fortunately, **1z** was reduced completely to give dihydroartemisinic acid (R)-**2z** with 98% yield and 99.8:0.2 dr, even at a 0.020 mol% catalyst loading (S/C = 5000), indicating its potential for industrial application (Fig. 3b). As a non-steroid anti-inflammatory drug, (S)-Ibuprofen ((S)-**2l**) could be obtained via asymmetric hydrogenation of **1l** using a catalyst loading of 0.10 mol% (1000 S/C) (Fig. 3c, 99% yield and 96% ee). (S)-Flurbiprofen ((S)-**2aa**) and (S)-Naproxen ((S)-**2ab**) could also be obtained with the same 95% ee at 50 and 100 S/C, respectively, due to poor solubility (Fig. 3d, e).

### The deuterium-labeling experiments

In order to explore a possible mechanism, deuterium-labelling experiments were conducted (Fig. 4). The asymmetric hydrogenation of **1f** was conducted in TFE under 30 bar of D$_2$. The product was obtained with >95% deuteration at the α-position of the carboxyl group and <5% deuteration at the methyl group (β-position of the carboxyl group). When CF$_3$CH$_2$OD was employed as the deuterated solvent, the product was obtained with 73% β-D and 7% α-D. These results suggest that the two added hydrogen atoms (α-H/β-H) of the products originate predominantly from H$_2$ and the protic solvent, respectively. This is different from our previous studies concerning α-substituted vinylphosphonates, in which the two added hydrogen atoms of the product originate from H$_2$[66].

### Mechanistic considerations

Combined with the results of the above deuterium-labelling experiments, we further investigated the reaction mechanism through DFT calculations (Fig. 5a, b). The heterolytic cleavage of hydrogen by the Ni complex generates a Ni-H complex, which then coordinates with the C=C bond of substrate **1a** to form intermediate **IM-1**. Next, the migratory insertion of Ni(II)-H to the vinyl group occurs, giving intermediates **IM-2S** and **IM-2R** via transition states **TS-1S** (1.78 kcal/mol) and **TS-1R** (0.97 kcal/mol), respectively. This step is reversible due to the low activation energy. Afterwards, one molecule of TFE solvent

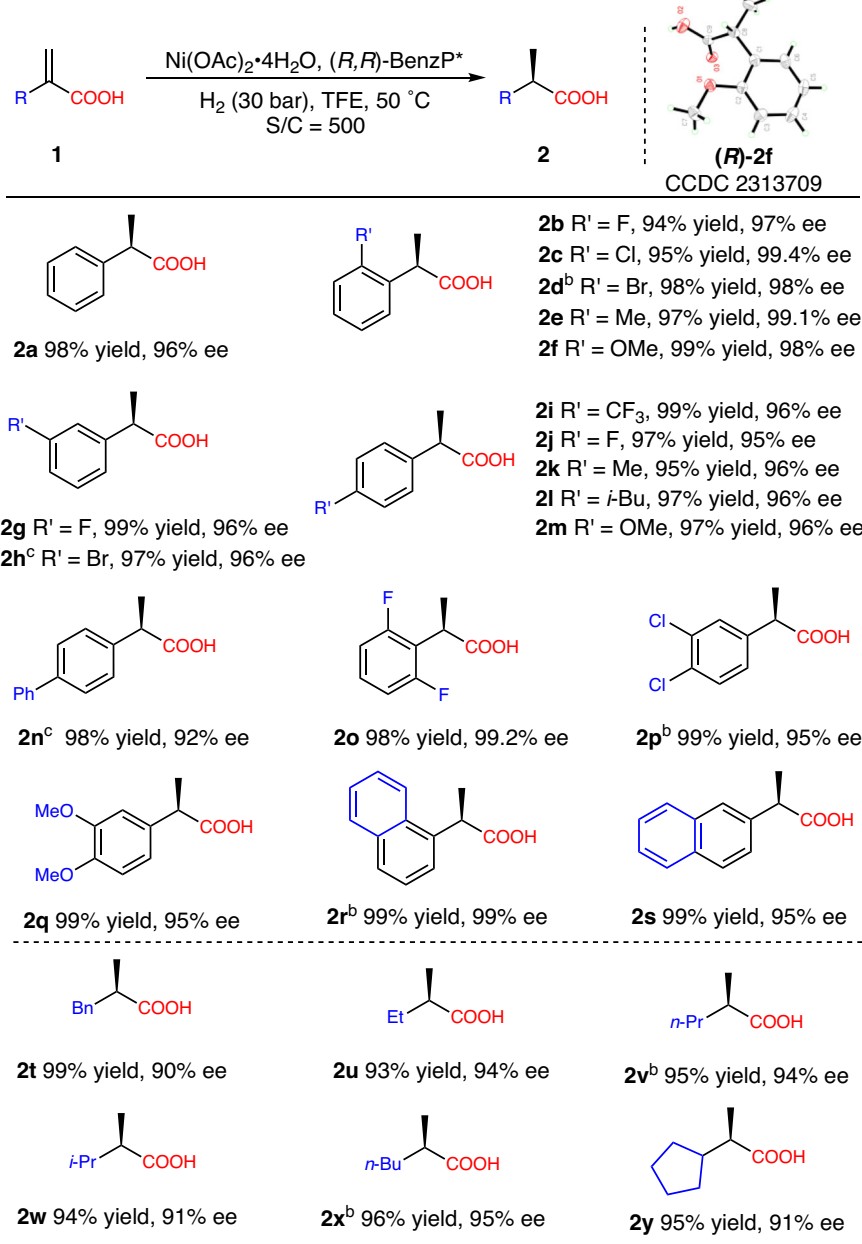

**Fig. 2 | Substrate scope[a].** Reaction conditions unless otherwise noted: **1** (0.50 mmol), Ni(OAc)₂·4H₂O/(R,R)-BenzP* (S/C = 500), H₂ (30 bar), TFE (2.0 mL), 50 °C, 24 h. [b]Conditions: **1** (0.50 mmol), Ni(OAc)₂·4H₂O/(R,R)-BenzP* (S/C = 250). [c]Conditions: **1** (0.20 mmol), Ni(OAc)₂·4H₂O /(R,R)-BenzP* (S/C = 100).

coordinates with **IM-2**, forming **IM-3S** and **IM-3R**. After the proto-nolysis of the C-Ni bond of **IM-3** via intramolecular proton transfer from the carboxylic acid group of the substrate via **TS-2**, the product **2a** is generated and the Ni complex is released. This is the rate-determining and stereo-determining step, and the $\Delta\Delta G^{\ddagger}$ (2.08 kcal/mol) corresponds to the favored configuration $R$ with 93% ee, match-ing the experimental data (Fig. 5b).

Next, reaction order studies were carried out to further verify the rate-determining step (Fig. 6a–c, also see Supplementary Tables 2–7 and Supplementary Figs. 5–7 for details). The results indicate that the reaction is first order with respect to the active nickel species or sub-strate and zero order with respect to H₂ pressure.

**Kinetic equation derivation**
Based on the above experimental and DFT computation results (Fig. 5), a series of kinetic equations could be derived as follows

(Fig. 6d). According to the above experimental results of reaction order studies (Fig. 6a–c), a kinetic equation for this reaction can be written as eq.1. As the generation of Ni-H complex is rapid due to zero order reaction with regards to H₂ pressure, this step should not be considered in the kinetic equation. Then, on account of the DFT result, there are two main elementary reactions in the hydrogenation (eq.2 and 3) in which the migratory insertion step (eq.2) is fast and reversible, while the intramolecular proton transfer step is slow and irreversible. These two elementary reactions perfectly satisfy the conditions required for the equilibrium hypothesis, so that eq.4 can be written. Subsequently, according to eq.3, the rate of generation of **2a** relies on **IM-2** and the solvent TFE, therefore eq.5 is obtained. Substituting eq.4 into eq.5 gives eq.6 after simplification, which is the same as eq.1. The same form of the kinetic equation derived from the DFT calculations and experimental results proves the rationality of our proposed mechanism.

**Fig. 3 | The study of catalyst efficiency and practical applications. a** A gram scale experiment of **1a** and the further transformation of the corresponding product **2a**. Conditions i: Ni(OAc)$_2$·4H$_2$O (0.050 mol%), (S,S)-BenzP* (0.010 mol%, S/C = 10000), H$_2$ (30 bar), TFE (12 mL), 60 °C, 3 days. **b** Asymmetric hydrogenation of artemisinic acid (**1z**) and further synthesis of artemisinin. Conditions ii: Ni(OAc)$_2$·4H$_2$O (0.20 mol%), (R,R)-BenzP* (0.020 mol%, S/C = 5000), H$_2$ (60 bar), TFE (12 mL), EtOAc (6.0 mL), 60 °C, 3 days. **c** Asymmetric hydrogenation to synthesize (S)-Ibuprofen. **d** Asymmetric hydrogenation to synthesize (S)-Flurbiprofen. **e** Asymmetric hydrogenation to synthesize (S)-Flurbiprofen.

**Fig. 4 | The deuterium-labeling experiments.** The deuterium-labeling asymmetric hydrogenations of **1f** was conducted using D$_2$ or CF$_3$CH$_2$OD.

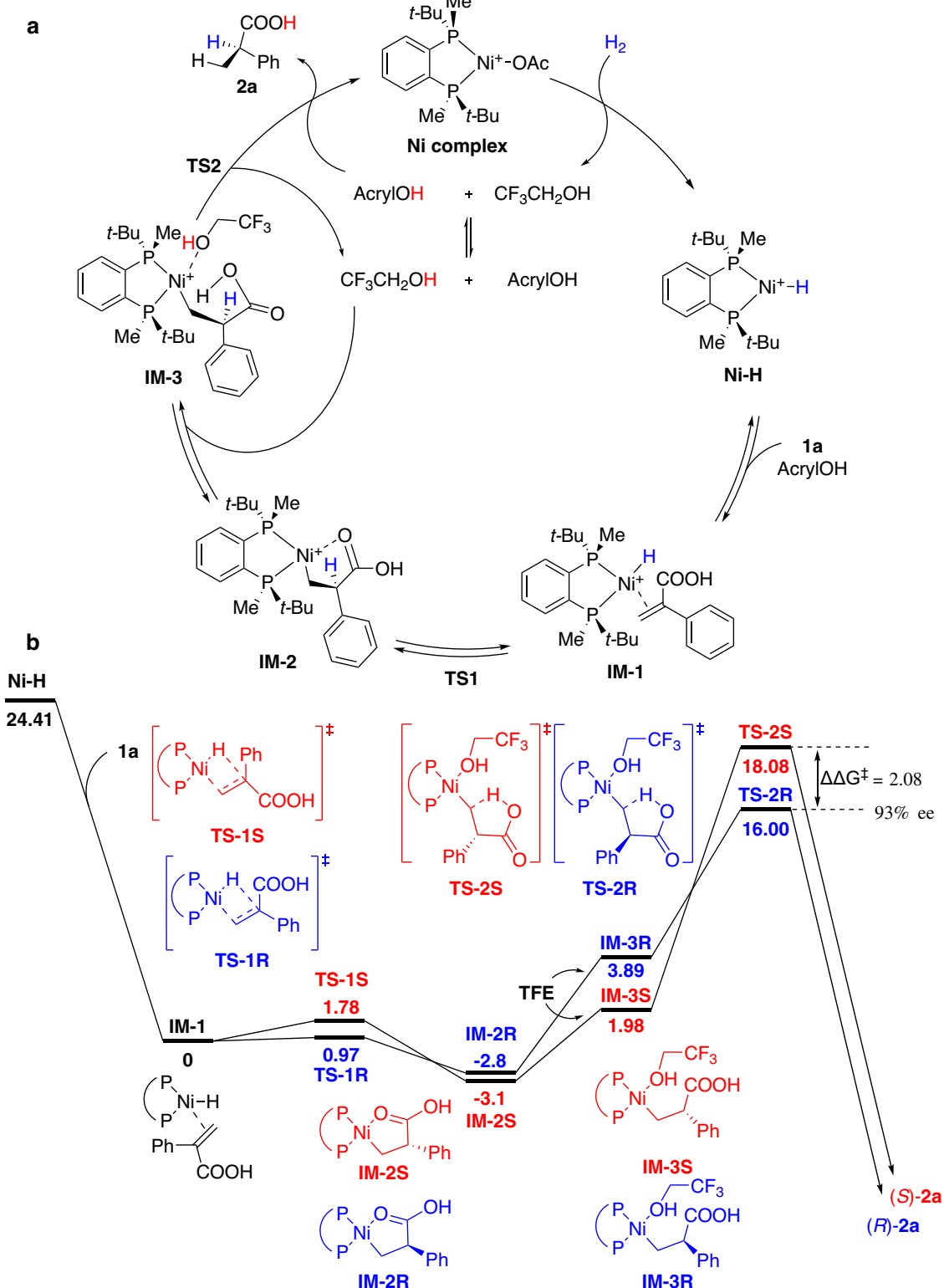

**Fig. 5 | Study on the catalytic mechanism. a** Proposed catalytic cycle. **b** DFT calculation of Ni-catalyzed asymmetric hydrogenation of **1a**.

## The Control Experiments

Subsequently, in order to further verify the effect of the carboxylic acid group in the catalytic cycle (**TS-2**), control experiments were conducted by adding acid (AcOH), bases (Na$_2$CO$_3$ and Et$_3$N) or using the corresponding ester substrate (**1ac**) accordingly (Fig. 7a–c). From Fig. 7a, it can be observed that almost no effect was seen by adding an extra 1.0 or 10.0 equivalents of acetic acid (51-52% conv. and 96% ee).

This result suggests that the reaction is not a traditional proton-dissociation mechanism, in which increasing the acidity of the reaction in a protic solvent increases the reaction rate[63,65]. When 1.0 equivalent of base was added, the reaction activity was reduced significantly (0.50 equiv. Na$_2$CO$_3$ or 1.0 equiv. Et$_3$N, 46% and 49% conv., respectively, Fig. 7b). As the amount of base increases (1.0 equiv. Na$_2$CO$_3$ or 2.0 equiv. Et$_3$N, Fig. 7b), no reaction occurs. This result suggests that our

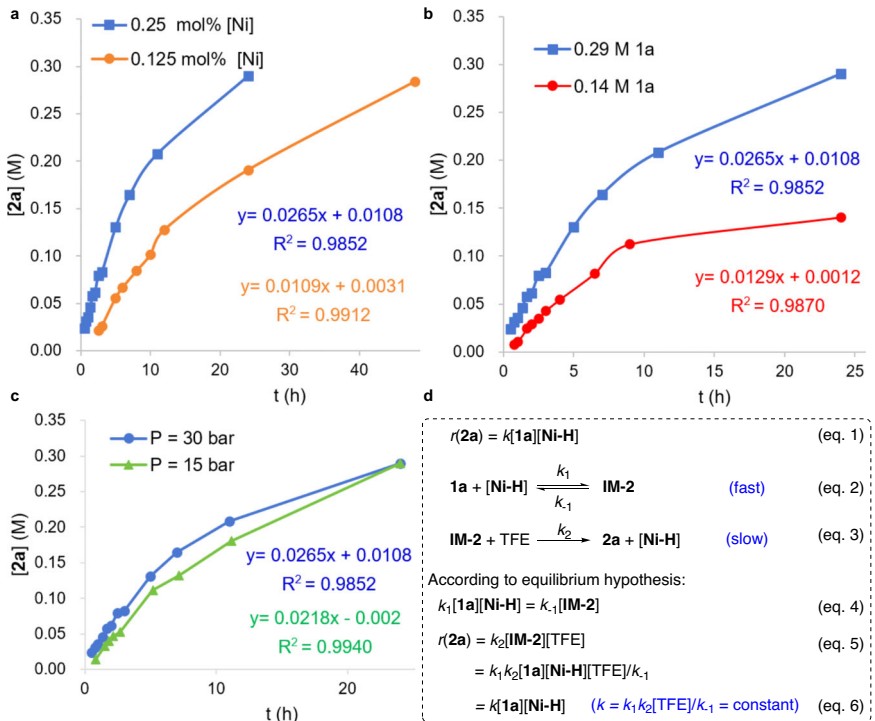

**Fig. 6 | The reaction order studies and kinetic equation derivation. a** The reaction order of catalyst. **b** The reaction order of substrate **1a**. **c** The reaction order of hydrogen pressure. **d** The process of kinetic equation derivation.

**Fig. 7 | The control experiments of the Ni-catalyzed asymmetric hydrogenation of 1a and 1ac. a** The control experiments by adding acid. **b** The control experiments by adding bases. **c** The control experiments by using substrate **1ac**.

reaction mechanism is also different from the previous reports, in which the substrate carboxylate anion coordinates with the metal center[30,69]. In addition, the hydrogenation of the esterified substrate **1ac** did not occur under standard hydrogenation conditions (<5%, Fig. 7c). These results clearly showed that the proton of the carboxylic acid group of the substrate plays a crucial role in the hydrogenation and is in agreement with the calculated catalytic cycle.

**Hirshfeld partition (IGMH) analysis**

In addition, we used an independent gradient model based on Hirshfeld partition (IGMH) analysis to provide the visualization of the secondary interactions between substrates and Ni catalysts species in **TS-2R** and **TS-2S** (see Supplementary Fig. 9 for details). Some C-H···H-C and C-H···O interactions are found in both transition states. However, these interactions are significantly weaker in **TS-2S** compared to **TS-2R**. Thus, this

IGMH analysis suggests that these weak interactions may participate in stabilizing the transition state and enhancing enantioselectivity.

## Discussion

In conclusion, an efficient earth-abundant metal nickel-catalyzed asymmetric hydrogenation of α-aryl and alkyl-substituted acrylic acids was developed. The corresponding chiral α-substituted propionic acids were obtained with excellent results (up to 99% yield, 99.4% ee, 10,000 S/C). Several acrylic acid drugs and drug intermediates were efficiently synthesized using this method. In particular, the key intermediate of Artemisinin, (*R*)-dihydroartemisinic acid, could be obtained with up to 99.8:0.2 dr and 5000 S/C. The mechanistic study suggested that the protonolysis of the C-Ni bond is the rate-determining step, which involves intramolecular proton transfer from the carboxylic acid group of the substrate.

## Methods

### General procedure for asymmetric hydrogenation of α-substituted acrylic acids

To a hydrogenation tube, Ni(OAc)$_2$·4H$_2$O (0.25 mg, 0.001 mmol), (*R,R*)-BenzP* (0.28 mg, 0.001 mmol) and the substrate **1** (S/C = 500) were added, and then the mixture was transferred to a nitrogen-filled glovebox. The degassed and anhydrous trifluoroethanol (TFE, 2.0 mL) was added. The reaction was performed with H$_2$ (30 bar) at 50 °C for 24 h. After carefully releasing hydrogen gas, the pure product is obtained by column chromatography (DCM/MeOH). The product was reacted with K$_2$CO$_3$/Me$_2$SO$_4$ or DMAP/DCC/aniline to afford the corresponding methyl ester or amide, whose enantiomeric excess was determined by HPLC with a chiral column.

## Data availability

The authors declare that the data supporting the findings of this study are available within the article and its Supplementary Information file. For the experimental procedures, and data of NMR and HPLC analysis, see Supplementary Methods and Charts in the Supplementary Information file. Source data of Cartesian coordinates of the optimized structures are provided in this paper. All data are available from the corresponding author upon request. The X-ray crystallographic coordinates for structures reported in this study have been deposited at the Cambridge Crystallographic Data Centre (CCDC), under deposition numbers CCDC 2313709 (***R*−2f**). These data can be obtained free of charge from The Cambridge Crystallographic Data Centre via www.ccdc.cam.ac.uk/data_request/cif. Source data are provided in this paper.

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

## Acknowledgements
We would like to thank the National Key R&D Program of China (Nos. 2023YFA1506400, W.Z. and J.C.; and 2023YFA1506700, D.L.) and the National Natural Science Foundation of China (Nos. 22361132533, W.Z.; 21991112, W.Z.) for financial support. We thank the Instrumental Analysis Center of SJTU for its characterization.

## Author contributions
B.L. and Z.W. conducted most of the synthetic experiments. Y.L. conducted the DFT computational study. H.W. and D.L. conducted some of the synthetic experiments. All authors discussed the results. J.C. and W.Z. designed the experiments and wrote the manuscript. W.Z. directed the project.

## Competing interests
The authors declare no competing interests.
