## [Peer Review File · Nature Communications]

Nickel-Catalyzed Asymmetric Hydrogenation for the Preparation of α -Substituted Propionic AcidsREVIEWER COMMENTS

Reviewer #1 (Remarks to the Author):

Zhang and coworkers describe an earth-abundant metal nickel catalyzed asymmetric hydrogenation of α -substituted acrylic acids with a wide range of substrates in this manuscript. The corresponding chiral products were obtained with excellent yields and enantioselectivities (up to 99% yield, 99.4% ee, and 10000 S/C). Especially, (R)-dihydroartemisinic acid product, which is the key intermediate of Artemisinin, could be achieved with up to 99.8:0.2 dr and 5000 S/C. In view of such excellent catalytic effect, the method can almost be applied to the industrial production of Artemisinin. The proposed mechanism is proved by deuterium-labeling experiments, control experiments and DFT calculation, which demonstrated that intramolecular proton transfer to break C-Ni bond is the key step in the reaction. In addition, this manuscript and supporting information are well-written. Thus, this manuscript should be of general interest for publication in Nature Communications once the following comments are addressed.

- 1) I found that only TFE worked significantly better than the other solvents. This asymmetric hydrogenation with high reactivity owned the potential in industry. If this method is applied to industrial production, can this solvent be replaced? Or a partial replacement?
- 2) In the second paragraph, 1988 should be 1987.
- 3) The authors show the catalytic results of a series of aryl and alkyl substrates. What are the catalytic results of heterocyclic substrates in this catalytic system?
- 4) Whether the challenging trisubstituted substrate is suitable for this system?
- 5) In the Methods part, please check the amount of solvent, which is 2 mL in standard conditions.

Reviewer #2 (Remarks to the Author):

The manuscript by Zhang and Chen reports an efficient asymmetric hydrogenation of α -aryl and alkyl substituted acrylic acids catalyzed by the earth-abundant metal nickel. This method yields chiral α -substituted propionic acids with outstanding results (up to 99% yield, 99.4% ee, 10,000 S/C). Various acrylic acid drugs and drug intermediates were efficiently synthesized using this approach. Notably, the key intermediate of Artemisinin, (R)-dihydroartemisinic acid, was obtained with up to 99.8:0.2 dr and 5000 S/C. Mechanistic studies suggest that the rate-determining step involves the protonolysis of the C-Ni bond, which includes intramolecular proton transfer from the substrate's carboxylic acid group. This work is suitable for publication in Nature Communications.

I have a few minor questions regarding this work:

1. It would be beneficial to observe a slightly greater variation of heteroaromatic substituted groups in Table 2.
2. Overall, the protocol demonstrates robustness, enabling the preparation of complex molecules with excellent results. Could the authors also discuss the limitations of the method? This information would be crucial for the further application of the protocol by drug discovery teams.

Reviewer #3 (Remarks to the Author):

In this manuscript, the authors present their results on the enantioselective hydrogenation of α -substituted acrylic acids in the presence of nickel catalysts.

The enantioselective hydrogenation of acrylic acids has been studied for a long time with noble metals (rhodium, ruthenium, iridium), but the use of 3d transition metal catalysts (cobalt, nickel) is much more recent. The authors should quote the recent review on this topic by Bruneau (*Tetrahedron* 2024, 151, 133793). The novelty in the present work lies in the use of nickel catalyst associated with a chiral diphosphine ligand (BenzP*), which gives high conversions and enantioselectivities for α -substituted acrylic acids. The very related results obtained by Huang and Hou on enantioselective hydrogenation of β,β -disubstituted acrylic acids in the presence of nickel catalyst must be cited as well.

The high yields, TONs, and enantioselectivities do not constitute the most important feature in this work (see in the review cited above, many excellent results have already been obtained), but the performance of nickel catalysis and the mechanistic studies with a 3d transition metal seems unprecedented and deserve publication in *Nature Commun.* It is experimentally shown that acetic acid has no favorable role in the catalytic cycle. Would it be reasonable to consider that the initial coordinated acetate could be rapidly replaced by acrylate and that acetic acid has no role in the catalytic cycle depicted in Fig. 4. With this hypothesis, AcOH could be replaced by AcryIOH in Fig. 4?

Reviewer #4 (Remarks to the Author):

In this manuscript, Zhang and co-authors describe an efficient protocol for the preparation of α -substituted propionic acids through earth-abundant metal Ni catalyzed asymmetric hydrogenations. Excellent yields (up to 99%) and ees (up to 99.4%) were provided. It is worth mentioning that the catalyst efficiency (10,000 S/C) is almost comparable to that of rare metal catalysts, which provides the possibility of large-scale production using earth-abundant catalysts. The method exhibits a broad scope of α -aryl and α -alkyl substituted acrylic acids. This is a good solution to the asymmetric hydrogenation of such terminal olefins with less steric hindrance, using a green and inexpensive catalytic system. The synthesis examples of several drug molecules and gram-scale reactions further prove the practicability of this method.

The reaction mechanism was studied in detail through a series of deuteration experiments, control experiments and reaction order studies. In addition, DFT calculation and kinetic equation derivation further confirm the rationality of the proposed mechanism. The new mode of intramolecular proton transfer is beneficial to improve the efficiency of the hydrogenation reaction. In view of the important applications of chiral α -substituted propionic acid products in bioactive compounds and organic synthesis, this paper can arouse the interest of researchers in chemistry, pharmacy, biology and other fields. Therefore, this referee believes that this manuscript is very suitable for publication on *Nat. Commun.* if the following minor revisions are completed.

1. In control experiments, the referee wants to know if the substrate 1ac is completely inactive or has low activity in the system. After all, 500 S/C generally means a very low catalyst loading.
2. There are several space and italics mistakes that need to be corrected. e.g. in Table 1,

“50 bar H2” should be “50 bar H₂”; in Table 2, “(R,R)-BenzP*” should be “(R,R)-BenzP^{*}”.

3. In the References, the format of uppercase/lowercase letters in the title should be uniform.

4. In the Supplementary Information, for nBu, i-Bu, eq, equiv., and other abbreviations in P4, p5 and p31, please pay attention to the uniform writing of italics and other formats.

5. In the Supplementary Information, the product ee values should also be listed in the HPLC chart section.

Thank you very much for sending us the valuable comments and suggestions made by the reviewers for our paper entitled “*Nickel-Catalyzed Asymmetric Hydrogenation for the Preparation of α -Substituted Propionic Acids*” (NCOMMS-24-19253). We have now revised the manuscript according to the reviewers’ comments and suggestions. The point-to-point responses are listed as follow:

Reviewer #1

Comment: Zhang and coworkers describe an earth-abundant metal nickel catalyzed asymmetric hydrogenation of α -substituted acrylic acids with a wide range of substrates in this manuscript. The corresponding chiral products were obtained with excellent yields and enantioselectivities (up to 99% yield, 99.4% ee, and 10000 S/C). Especially, (*R*)-dihydroartemisinic acid product, which is the key intermediate of Artemisinin, could be achieved with up to 99.8:0.2 dr and 5000 S/C. In view of such excellent catalytic effect, the method can almost be applied to the industrial production of Artemisinin. The proposed mechanism is proved by deuterium-labeling experiments, control experiments and DFT calculation, which demonstrated that intramolecular proton transfer to break C-Ni bond is the key step in the reaction. In addition, this manuscript and supporting information are well-written. Thus, this manuscript should be of general interest for publication in Nature Communications once the following comments are addressed.

Our response:

We appreciate the reviewer’s kind comment.

Suggestion 1: I found that only TFE worked significantly better than the other solvents. This asymmetric hydrogenation with high reactivity owned the potential in industry. If this method is applied to industrial production, can this solvent be replaced? Or a partial replacement?

Our response:

We appreciate the reviewer’s kind suggestion. When several other polar and non-polar solvents (MeOH, EtOH, *i*-PrOH, toluene, THF and AcOH) were tested, the desired products were obtained with less than 10% conversions (entries 2-6, see Table R1 and S1). We have further optimized the solvents to improve the results. First, neat AcOH was selected as the solvent and the product was obtained in 23% yield and 95% ee (entry 7). Next, we investigated mixed solvent systems (entries 8-11). In the mixtures (2/1, v/v) of TFE/MeOH, TFE/EtOH, TFE/EA, the conversions are between 17-28%. To our delight, when using a TFE/AcOH mixture (2/1, v/v), the reaction proceeded well and gave the product with 99% conversion and 95% ee (entry 11). When the TFE/AcOH ratio was reduced to 1/2, the reaction still proceeded smoothly. When TFE/AcOH was further reduced to 1/10 and 1/20, the conversions were 81% and 57%, respectively. Therefore, although TFE is a necessary solvent for the best results, some alternative mixed solvents are also appropriate for further optimization.

Table R1. Additional investigation of the reaction solvents.^a

Entry	Solvent	Conv. (%)	Ee (%)
1	TFE	>99	96
2	MeOH	7%	-
3	EtOH	8%	-
4	i -PrOH	-	-
5	toluene	-	-
6	THF	-	-
7	AcOH	23	95
8	TFE:MeOH 2:1	19	94
9	TFE:EtOH 2:1	17	93
10	TFE:EA 2:1	28	96
11	TFE:AcOH 2:1	>99	95
12	TFE:AcOH 1:2	>99	95
13	TFE:AcOH 1:10	81	95
14	TFE:AcOH 1:20	57	95

^a Reaction condition: **1a** (0.2 mmol), Ni(OAc)₂·4H₂O (1 mol%, S/C = 100), (*R,R*)-BenzP* (1 mol%), solvent (1.0 mL), H₂ (30 bar), 50 °C, 24 h. The conversions were detected by ¹H NMR spectra. The product was reacted to afford the corresponding methyl ester or amide, whose enantiomeric excess was determined by HPLC with a chiral column.

Suggestion 2: In the second paragraph, 1988 should be 1987.

Our response:

We appreciate the reviewer's kind suggestion. The "1988" in the second paragraph has been corrected to "1987" in the revised manuscript.

Suggestion 3: The authors show the catalytic results of a series of aryl and alkyl substrates. What are the catalytic results of heterocyclic substrates in this catalytic system?

Our response:

We appreciate the thoughtful suggestion from the reviewer. In response to the feedback provided by reviewer #1 and #2, we also explored heterocyclic substrates. However, compared to alkyl and aryl substrates, heterocyclic substrates or intermediates proved to be highly unstable. This instability likely explains the limited research on the asymmetric hydrogenation of such heterocyclic carboxylic acid substrates in the past. We attempted various synthesis methods for different heterocyclic substrates, but ultimately only achieved a low yield with the pyridine-substituted substrate. This pyridine substrate was tested in the asymmetric hydrogenation system, yielding a conversion of 18% (see Fig. R1). However, measuring its ee value proved impossible due to the generation of numerous impurities during the conversion to the corresponding ester. The catalytic results have been included in the revised Supporting Information.

Fig. S1 The heterocyclic substrates

Suggestion 4: Whether the challenging trisubstituted substrate is suitable for this system?

Our response:

We appreciate the reviewer's kind suggestion. According to the reviewer's suggestion, trisubstituted substrates were also studied in the catalytic system. Four different types (alkyl-alkyl, aryl-alkyl, aryl-aryl and alkyl-aryl) of E-trisubstituted substrates were synthesized and studied, but only the alkyl-aryl substrate gave 33% conversion and 87% ee. This may indicate that the catalytic system is more suitable for disubstituted carboxylic acid substrates (Fig. R2). The catalytic results have been added to the revised Supporting Information.

Fig. R2 The trisubstituted substrates

Suggestion 5: In the Methods part, please check the amount of solvent, which is 2 mL in standard conditions.

Our response:

We appreciate the reviewer's kind suggestion. Indeed, we used 2 mL solvent in the standard conditions and have corrected the description in the revised manuscript.

Reviewer #2

Comment: The manuscript by Zhang and Chen reports an efficient asymmetric hydrogenation of α -aryl and alkyl substituted acrylic acids catalyzed by the earth-abundant metal nickel. This method yields chiral α -substituted propionic acids with outstanding results (up to 99% yield, 99.4% ee, 10,000 S/C). Various acrylic acid drugs and drug intermediates were efficiently synthesized using this approach. Notably, the key intermediate of Artemisinin, (*R*)-dihydroartemisinic acid, was obtained with up to 99.8:0.2 dr and 5000 S/C. Mechanistic studies suggest that the rate-determining step involves the protonolysis of the C-Ni bond, which includes intramolecular proton transfer from the substrate's carboxylic acid group. This work is suitable for publication in Nature Communications.

Our response:

We appreciate the reviewer's kind comment.

Suggestion 1: It would be beneficial to observe a slightly greater variation of heteroaromatic substituted groups in Table 2.

Our response:

We appreciate the thoughtful suggestion from the reviewer. In response to the feedback provided by reviewer #1 and #2, we also explored heteroaromatic substrates. However, compared to alkyl and aryl substrates, heteroaromatic substrates or intermediates proved to be highly unstable. This instability likely explains the limited research on the asymmetric hydrogenation of such heteroaromatic carboxylic acid substrates in the past. We attempted various synthesis methods for different heteroaromatic substrates, but ultimately only achieved a low yield with the pyridine-substituted substrate. This pyridine substrate was tested in the asymmetric hydrogenation system, yielding a conversion of 18% (see Fig. R1). However, measuring its ee value proved impossible due

to the generation of numerous impurities during the conversion to the corresponding ester. The catalytic results have been included in the revised Supporting Information.

Fig. S1 The heteroaromatic substrates

Suggestion 2: Overall, the protocol demonstrates robustness, enabling the preparation of complex molecules with excellent results. Could the authors also discuss the limitations of the method? This information would be crucial for the further application of the protocol by drug discovery teams.

Our response:

We appreciate the reviewer's kind suggestion. Although this catalytic system has achieved good results in a series of substrates and several applications are shown, there are still some limitations. For example, the heteroaromatic substrates, which are not easy to synthesize, did not provide good catalytic results in this system. This catalytic system is also not suitable for trisubstituted carboxylic acid substrates. Furthermore, while there are alternative mixed solvents available, only TFE solvent demonstrated superior performance. However, the selectivity of other solvents remains limited. The catalytic results have been included in the revised Supporting Information.

Reviewer #3

Comment: In this manuscript, the authors present their results on the enantioselective hydrogenation of α -substituted acrylic acids in the presence of nickel catalysts.

Our response:

We appreciate the reviewer's kind comment.

Suggestion 1: The enantioselective hydrogenation of acrylic acids has been studied for a long time with noble metals (rhodium, ruthenium, iridium), but the use of 3d transition metal catalysts (cobalt, nickel) is much more recent. The authors should quote the recent review on this topic by Bruneau (Tetrahedron 2024, 151, 133793). The novelty in the present work lies in the use of nickel catalyst associated with a chiral diphosphine ligand (BenzP*), which gives high conversions and enantioselectivities for α -substituted acrylic acids. The very related results obtained by Huang and Hou on enantioselective hydrogenation of β,β -disubstituted acrylic acids in the presence of nickel catalyst must be cited as well.

Our response:

We appreciate the reviewer's kind suggestion. The two relevant articles have been cited in the revised manuscript as references 19 and 70, respectively.

Suggestion 2: The high yields, TONs, and enantioselectivities do not constitute the most important feature in this work (see in the review cited above, many excellent results have already been obtained), but the performance of nickel catalysis and the mechanistic studies with a 3d transition metal seems unprecedented and deserve publication in Nature Commun. It is experimentally shown that acetic acid has no favorable role in the catalytic cycle. Would it be reasonable to consider that the initial coordinated acetate could be rapidly replaced by acrylate and that acetic acid has no role in the catalytic cycle depicted in Fig. 4. With this hypothesis, AcOH could be replaced by AcrylOH in Fig. 4?

Our response:

We appreciate the reviewer's kind suggestion. It is very reasonable that the initial coordinated acetate can be rapidly replaced by acrylate. We have changed AcOH to AcrylOH in Fig. 4 in the revised manuscript.

Reviewer #4

Comment: In this manuscript, Zhang and co-authors describe an efficient protocol for the preparation of α -substituted propionic acids through earth-abundant metal Ni catalyzed asymmetric hydrogenations. Excellent yields (up to 99%) and ees (up to 99.4%) were provided. It is worth mentioning that the catalyst efficiency (10,000 S/C) is almost comparable to that of rare metal catalysts, which provides the possibility of large-scale production using earth-abundant catalysts. The method exhibits a broad scope of α -aryl and α -alkyl substituted acrylic acids. This is a good solution to the asymmetric hydrogenation of such terminal olefins with less steric hindrance, using a green and inexpensive catalytic system. The synthesis examples of several drug molecules and gram-scale reactions further prove the practicability of this method.

The reaction mechanism was studied in detail through a series of deuteration experiments, control experiments and reaction order studies. In addition, DFT calculation and kinetic equation derivation further confirm the rationality of the proposed mechanism. The new mode of intramolecular proton

transfer is beneficial to improve the efficiency of the hydrogenation reaction. In view of the important applications of chiral α -substituted propionic acid products in bioactive compounds and organic synthesis, this paper can arouse the interest of researchers in chemistry, pharmacy, biology and other fields. Therefore, this referee believes that this manuscript is very suitable for publication on Nat. Commun. if the following minor revisions are completed.

Our response:

We appreciate the reviewer's kind comment.

Suggestion 1: In control experiments, the referee wants to know if the substrate **1ac** is completely inactive or has low activity in the system. After all, 500 S/C generally means a very low catalyst loading.

Our response:

We appreciate the reviewer's kind suggestion. The hydrogenation of the esterified substrate **1ac** did not occur under the standard hydrogenation conditions (<5%, S/C = 500, Fig. R3). Reducing the S/C or adding acid simultaneously increased the conversion (22% and 63%) of **1ac** and provided slightly lower enantioselectivities of 91%. These experiments show that esterified substrate **1ac** has low activity and good enantioselectivity in this system. The results have been added to the revised Supporting Information.

Fig. S3 Nickel catalyzed asymmetric hydrogenation of **1ac**

Suggestion 2: There are several space and italics mistakes that need to be corrected. e.g. in Table 1, “50 bar H₂” should be “50 bar H₂”; in Table 2, “(R,R)-BenzP*” should be “(R,R)-BenzP*”.

Our response:

We appreciate the reviewer's kind suggestion. The corresponding space and italics mistakes have been corrected in Table 1 and Table 2 in the revised manuscript.

Suggestion 3: In the References, the format of uppercase/lowercase letters in the title should be uniform.

Our response:

We appreciate the reviewer's kind suggestion. The format of uppercase/lowercase letters in each references' title have been corrected uniformly in the revised manuscript.

Suggestion 4: In the Supplementary Information, for nBu, i-Bu, eq, equiv., and other abbreviations in P4, p5 and p31, please pay attention to the uniform writing of italics and other formats.

Our response:

We appreciate the reviewer's kind suggestion. We have checked the Supplementary Information carefully and corrected all the inconsistent writing of italics and other formats in the revised

Supplementary Information.

Suggestion 5: In the Supplementary Information, the product ee values should also be listed in the HPLC chart section.

Our response:

We appreciate the reviewer's kind suggestion. The product ee values have been listed in the HPLC chart section in the revised Supplementary Information.

In addition, when we revised the manuscript, Dr. Liu conducted some of the synthetic experiments and made some helpful suggestions. Accordingly, he is listed as a co-author, which has been approved by all of the authors. We have highlighted the changes made during revision by giving the text a yellow background. We hope that the revised manuscript satisfies the criteria as an article for publication in *Nature Communication*. Please feel free to contact us if you have any questions. Once again, we would like to thank you and the referees for providing comments and suggestions for our manuscript.

Yours sincerely,

Prof. Wanbin Zhang
School of Chemistry and Chemical Engineering
Shanghai Jiao Tong University
800 Dongchuan Road, Shanghai 200240, P. R. China
Phone: +86-21-54743265; Fax: +86-21-54743265
E-mail: wanbin@sjtu.edu.cn
Homepage: wanbin.sjtu.edu.cn

REVIEWERS' COMMENTS

Reviewer #1 (Remarks to the Author):

This revised manuscript can be accepted and published as the the current form.

Reviewer #2 (Remarks to the Author):

I have reviewed the original submission of this manuscript. The authors carefully revised it according to the comments by reviewers. Now, the revised version is suitable for publication in Nature Communications.

Reviewer #4 (Remarks to the Author):

The authors have resolved all the confusion, and I recommend accepting it directly.

Thank you very much for sending us the valuable comments and suggestions made by the reviewers for our paper entitled “*Nickel-Catalyzed Asymmetric Hydrogenation for the Preparation of α -Substituted Propionic Acids*” (NCOMMS-24-19253A). We have now revised the manuscript according to the reviewers’ comments and suggestions. The point-to-point responses are listed as follow:

Reviewer #1

Comment: This revised manuscript can be accepted and published as the the current form.

Our response:

We very much appreciate the reviewer’s kind comment on our revisions very much.

Reviewer #2

Comment: I have reviewed the original submission of this manuscript. The authors carefully revised it according to the comments by reviewers. Now, the revised version is suitable for publication in Nature Communications.

Our response:

We very much appreciate the reviewer’s kind comment on our revisions very much.

Reviewer #4

Comment: The authors have resolved all the confusion, and I recommend accepting it directly.

Our response:

We very much appreciate the reviewer’s kind comment on our revisions very much.

In addition, the editorial requests have been addressed in the attached Author Checklist. We hope that the revised manuscript satisfies the criteria as an article for publication in *Nature Communication*. Please feel free to contact us if you have any questions. Once again, we would like to thank you and the referees for providing comments and suggestions for our manuscript.

Yours sincerely,

Prof. Wanbin Zhang
School of Chemistry and Chemical Engineering
Shanghai Jiao Tong University
800 Dongchuan Road, Shanghai 200240, P. R. China
Phone: +86-21-54743265; Fax: +86-21-54743265
E-mail: wanbin@sjtu.edu.cn
Homepage: wanbin.sjtu.edu.cn